# Bayesian Sampling Using Stochastic Gradient Thermostats

**Nan Ding**[*]
Google Inc.
dingnan@google.com

**Youhan Fang**[*]
Purdue University
yfang@cs.purdue.edu

**Ryan Babbush**
Google Inc.
babbush@google.com

**Changyou Chen**
Duke University
cchangyou@gmail.com

**Robert D. Skeel**
Purdue University
skeel@cs.purdue.edu

**Hartmut Neven**
Google Inc.
neven@google.com

## Abstract

Dynamics-based sampling methods, such as Hybrid Monte Carlo (HMC) and Langevin dynamics (LD), are commonly used to sample target distributions. Recently, such approaches have been combined with stochastic gradient techniques to increase sampling efficiency when dealing with large datasets. An outstanding problem with this approach is that the stochastic gradient introduces an unknown amount of noise which can prevent proper sampling after discretization. To remedy this problem, we show that one can leverage a small number of additional variables to stabilize momentum fluctuations induced by the unknown noise. Our method is inspired by the idea of a thermostat in statistical physics and is justified by a general theory.

## 1 Introduction

The generation of random samples from a posterior distribution is a pervasive problem in Bayesian statistics which has many important applications in machine learning. The Markov Chain Monte Carlo method (MCMC), proposed by Metropolis et al.[16], generates unbiased samples from a desired distribution when the density function is known up to a normalizing constant. However, traditional MCMC methods are based on random walk proposals which lead to highly correlated samples. On the other hand, dynamics-based sampling methods, *e.g.* Hybrid Monte Carlo (HMC) [6, 10], avoid this high degree of correlation by combining dynamic systems with the Metropolis step. The dynamic system uses information from the gradient of the log density to reduce the random walk effect, and the Metropolis step serves as a correction of the discretization error introduced by the numerical integration of the dynamic systems.

The computational cost of HMC methods depends primarily on the gradient evaluation. In many machine learning problems, expensive gradient computations are a consequence of working with extremely large datasets. In such scenarios, methods based on stochastic gradients have been very successful. A stochastic gradient uses the gradient obtained from a random subset of the data to approximate the true gradient. This idea was first used in optimization [9, 19] but was recently adapted for sampling methods based on stochastic differential equations (SDEs) such as Brownian dynamics [1, 18, 24] and Langevin dynamics [5].

Due to discretization, stochastic gradients introduce an unknown amount of noise into the dynamic system. Existing methods sample correctly only when the step size is small or when a good estimate of the noise is available. In this paper, we propose a method based on SDEs that self-adapts to the

---

[*] indicates equal contribution.

unknown noise with the help of a small number of additional variables. This allows for the use of larger discretization step, smaller diffusion factor, or smaller minibatch to improve the sampling efficiency without sacrificing accuracy.

From the statistical physics perspective, all these dynamics-based sampling methods are approaches that use dynamics to approximate a canonical ensemble [23]. In a canonical ensemble, the distribution of the states follows the canonical distribution which corresponds to the target posterior distribution of interests. In attemping to sample from the canonical ensemble, existing methods have neglected the condition that, the system temperature must remain near a target temperature (Eq.(4) of Sec. 3). When this requirement is ignored, noise introduced by stochastic gradients may drive the system temperature away from the target temperature and cause inaccurate sampling. The additional variables in our method essentially play the role of a thermostat which controls the temperature and, as a consequence, handles the unknown noise. This approach can also be found by following a general recipe which helps designing dynamic systems that produce correct samples.

The rest of the paper is organized as follows. Section 2 briefly reviews the related background. Section 3 proposes the stochastic gradient Nosé-Hoover thermostat method which maintains the canonical ensemble. Section 4 presents the general recipe for finding proper SDEs and mathematically shows that the proposed method produces samples from the correct target distribution. Section 5 compares our method with previous methods on synthetic and real world machine learning applications. The paper is concluded in Section 6.

## 2   Background

Our objective is to generate random samples from the posterior probability density $p(\boldsymbol{\theta}\,|\,\mathbf{X}) \propto p(\mathbf{X}\,|\boldsymbol{\theta})p(\boldsymbol{\theta})$, where $\boldsymbol{\theta}$ represents an $n$-dim parameter vector and $\mathbf{X}$ represents data. The canonical form is $p(\boldsymbol{\theta}\,|\,\mathbf{X}) = (1/Z)\exp(-U(\boldsymbol{\theta}))$ where $U(\boldsymbol{\theta}) = -\log p(\mathbf{X}\,|\boldsymbol{\theta}) - \log p(\boldsymbol{\theta})$ is referred to as the potential energy and $Z$ is the normalizing constant. Here, we briefly review a few dynamics-based sampling methods, including HMC, LD, stochastic gradient LD (SGLD) [24], and stochastic gradient HMC (SGHMC) [5], while relegating a more comprehensive review to Appendix A.

HMC [17] works in an extended space $\Gamma = (\boldsymbol{\theta}, \mathbf{p})$, where $\boldsymbol{\theta}$ and $\mathbf{p}$ simulate the positions and the momenta of particles in a system. Although some works, e.g. [7, 8], make use of variable mass, we assume that all particles have unit constant mass (i.e. $m_i = 1$). The joint density of $\boldsymbol{\theta}$ and $\mathbf{p}$ can be written as $\rho(\boldsymbol{\theta}, \mathbf{p}) \propto \exp(-H(\boldsymbol{\theta}, \mathbf{p}))$, where $H(\boldsymbol{\theta}, \mathbf{p}) = U(\boldsymbol{\theta}) + K(\mathbf{p})$ is called the Hamiltonian (the total energy). $U(\boldsymbol{\theta})$ is called the potential energy and $K(\mathbf{p}) = \mathbf{p}^\top \mathbf{p}/2$ is called the kinetic energy. Note that $\mathbf{p}$ has standard normal distribution. The force on the system is defined as $\mathbf{f}(\boldsymbol{\theta}) = -\nabla U(\boldsymbol{\theta})$. It can be shown that the Hamiltonian dynamics

$$d\boldsymbol{\theta} = \mathbf{p}\,dt, \quad d\,\mathbf{p} = \mathbf{f}(\boldsymbol{\theta})dt,$$

maintain a constant total energy [17]. In each step of the HMC algorithm, one first randomizes $\mathbf{p}$ according to the standard normal distribution; then evolves $(\boldsymbol{\theta}, \mathbf{p})$ according to the Hamiltonian dynamics (solved by numerical integrators); and finally uses the Metropolis step to correct the discretization error.

Langevin dynamics (with diffusion factor $A$) are described by the following SDE,

$$d\boldsymbol{\theta} = \mathbf{p}\,dt, \quad d\,\mathbf{p} = \mathbf{f}(\boldsymbol{\theta})dt - A\,\mathbf{p}\,dt + \sqrt{2A}d\,\mathbf{W}, \tag{1}$$

where $\mathbf{W}$ is $n$ independent Wiener processes (see Appendix A), and $d\,\mathbf{W}$ can be informally written as $\mathcal{N}(0, \mathbf{I}\,dt)$ or simply $\mathcal{N}(0, dt)$ as in [5]. Brownian dynamics

$$d\boldsymbol{\theta} = \mathbf{f}(\boldsymbol{\theta})dt + \mathcal{N}(0, 2dt)$$

is obtained from Langevin dynamics by rescaling time $t \leftarrow At$ and letting $A \to \infty$, i.e., on long time scales inertia effects can be neglected [11]. When the size of the dataset is big, the computation of the gradient of $-\log p(\mathbf{X}\,|\boldsymbol{\theta}) = -\sum_{i=1}^{N} \log p(\mathbf{x}_i\,|\boldsymbol{\theta})$ can be very expensive. In such situations, one could use the likelihood of a random subset of the data $\mathbf{x}_i$'s to approximate the true likelihood,

$$\tilde{U}(\boldsymbol{\theta}) = -\frac{N}{\tilde{N}}\sum_{i=1}^{\tilde{N}} \log p(\mathbf{x}_{(i)}\,|\boldsymbol{\theta}) - \log p(\boldsymbol{\theta}), \tag{2}$$

where $\{\mathbf{x}_{(i)}\}$ represents a random subset of $\{\mathbf{x}_i\}$ and $\tilde{N} \ll N$. Define the stochastic force $\tilde{\mathbf{f}}(\boldsymbol{\theta}) = -\nabla \tilde{U}(\boldsymbol{\theta})$. The SGLD algorithm [24] uses $\tilde{\mathbf{f}}(\boldsymbol{\theta})$ and the Brownian dynamics to generate samples,

$$d\boldsymbol{\theta} = \tilde{\mathbf{f}}(\boldsymbol{\theta})dt + \mathcal{N}(0, 2dt).$$

In [5], the stochastic force with a discretization step $h$ is approximated as $h\tilde{\mathbf{f}}(\boldsymbol{\theta}) \simeq h\,\mathbf{f}(\boldsymbol{\theta}) + \mathcal{N}(0, 2h\,\mathbf{B}(\boldsymbol{\theta}))$ (note that the argument is not rigorous and that other significant artifacts of discretization may have been neglected). The SGHMC algorithm uses a modified LD,

$$d\boldsymbol{\theta} = \mathbf{p}\,dt, \quad d\,\mathbf{p} = \tilde{\mathbf{f}}(\boldsymbol{\theta})dt - A\,\mathbf{p}\,dt + \mathcal{N}(0, 2(A\,\mathbf{I} - \hat{\mathbf{B}}(\boldsymbol{\theta}))dt), \tag{3}$$

where $\hat{\mathbf{B}}(\boldsymbol{\theta})$ is intended to offset $\mathbf{B}(\boldsymbol{\theta})$, the noise from the stochastic force.

However, $\hat{\mathbf{B}}(\boldsymbol{\theta})$ is hard to estimate in practice and cannot be omitted when the discretization step $h$ is not small enough. Since poor estimation of $\hat{\mathbf{B}}(\boldsymbol{\theta})$ may lead to inaccurate sampling, we attempt to find a dynamic system which is able to adaptively fit to the noise without explicit estimation. The intuition comes from the practice of sampling a canonical ensemble in statistical physics.

The Metropolis step in SDE-based samplers with stochastic gradients is sometimes omitted on large datasets, because the evaluation of the potential energy requires using the entire dataset which cancels the benefit of using stochastic gradients. There is some recent work [2, 3, 14] which attempts to estimate the Metropolis step using partial data. Although an interesting direction for future work, in this paper we do not consider applying Metropolis step in conjunction with stochastic gradients.

# 3 Stochastic Gradient Thermostats

In statistical physics, a canonical ensemble represents the possible states of a system in thermal equilibrium with a heat bath at fixed temperature $T$ [23]. The probability of the states in a canonical ensemble follows the canonical distribution $\rho(\boldsymbol{\theta}, \mathbf{p}) \propto \exp(-H(\boldsymbol{\theta}, \mathbf{p})/(k_B T))$, where $k_B$ is the Boltzmann constant. A critical characteristic of the canonical ensemble is that the system temperature, defined as the mean kinetic energy, satisfies the following thermal equilibrium condition,

$$\frac{k_B T}{2} = \frac{1}{n}\,\mathbb{E}[K(\mathbf{p})], \quad \text{or equivalently,} \quad k_B T = \frac{1}{n}\,\mathbb{E}[\mathbf{p}^\top \mathbf{p}]. \tag{4}$$

All dynamics-based sampling methods approximate the canonical ensemble to generate samples. In Bayesian statistics, $n$ is the dimension of $\boldsymbol{\theta}$, and $k_B T = 1$ so that $\rho(\boldsymbol{\theta}, \mathbf{p}) \propto \exp(-H(\boldsymbol{\theta}, \mathbf{p}))$ and more importantly $\rho_\theta(\boldsymbol{\theta}) \propto \exp(-U(\boldsymbol{\theta}))$. However, one key fact that was overlooked in previous methods, is that the dynamics that correctly simulate the canonical ensemble must maintain the thermal equilibrium condition (4). Besides its physical meaning, the condition is necessary for $\mathbf{p}$ being distributed as its marginal canonical distribution $\rho_p(\mathbf{p}) \propto \exp(-K(\mathbf{p}))$.

It can be verified that ordinary HMC and LD (1) with true force both maintain (4). However, after combination with the stochastic force $\tilde{\mathbf{f}}(\boldsymbol{\theta})$, the dynamics (3) may drift away from thermal equilibrium if $\hat{\mathbf{B}}(\boldsymbol{\theta})$ is poorly estimated. Therefore, to generate correct samples, one needs to introduce a proper *thermostat*, which adaptively controls the mean kinetic energy. To this end, we introduce an additional variable $\xi$, and use the following dynamics (with diffusion factor $A$ and $k_B T = 1$),

$$d\boldsymbol{\theta} = \mathbf{p}\,dt, \quad d\,\mathbf{p} = \tilde{\mathbf{f}}(\boldsymbol{\theta})dt - \xi\,\mathbf{p}\,dt + \sqrt{2A}\,\mathcal{N}(0, dt), \tag{5}$$

$$d\xi = (\frac{1}{n}\,\mathbf{p}^\top \mathbf{p} - 1)dt. \tag{6}$$

Intuitively, if the mean kinetic energy is higher than 1/2, then $\xi$ gets bigger and $\mathbf{p}$ experiences more friction in (5); on the other hand, if the mean kinetic energy is lower, then $\xi$ gets smaller and $\mathbf{p}$ experiences less friction. Because (6) appears to be the same as the Nosé-Hoover thermostat [13] in statistical physics, we call our method stochastic gradient Nosé-Hoover thermostat (SGNHT, Algorithm 1). In Section 4, we will show that (6) is a simplified version of a more general SGNHT method that is able to handle high dimensional non-isotropic noise from $\tilde{\mathbf{f}}$. But before that, let us first look at a 1-D illustration of the SGNHT sampling in the presence of unknown noise.

**Algorithm 1:** Stochastic Gradient Nosé-Hoover Thermostat

---

**Input**: Parameters $h$, $A$.
Initialize $\boldsymbol{\theta}_{(0)} \in \mathbf{R}^n$, $\mathbf{p}_{(0)} \sim \mathcal{N}(0, \mathbf{I})$, and $\xi_{(0)} = A$ ;
**for** $t = 1, 2, \dots$ **do**

    Evaluate $\nabla \tilde{U}(\boldsymbol{\theta}_{(t-1)})$ from (2) ;

    $\mathbf{p}_{(t)} = \mathbf{p}_{(t-1)} - \xi_{(t-1)} \, \mathbf{p}_{(t-1)} \, h - \nabla \tilde{U}(\boldsymbol{\theta}_{(t-1)}) h + \sqrt{2A} \, \mathcal{N}(0, h)$;

    $\boldsymbol{\theta}_{(t)} = \boldsymbol{\theta}_{(t-1)} + \mathbf{p}_{(t)} \, h$;

    $\xi_{(t)} = \xi_{(t-1)} + (\frac{1}{n} \, \mathbf{p}_{(t)}^{\top} \, \mathbf{p}_{(t)} - 1) h$;

**end**

---

**Illustrations of a Double-well Potential**  To illustrate that the adaptive update (6) is able to control the mean kinetic energy, and more importantly, produce correct sampling with unknown noise on the gradient, we consider the following double-well potential,

$$U(\theta) = (\theta + 4)(\theta + 1)(\theta - 1)(\theta - 3)/14 + 0.5.$$

The target distribution is $\rho(\theta) \propto \exp(-U(\theta))$. To simulate the unknown noise, we let $\nabla \tilde{U}(\theta) h = \nabla U(\theta) h + \mathcal{N}(0, 2Bh)$, where $h = 0.01$ and $B = 1$. In the interest of clarity we did not inject additional noise other than the noise from $\nabla \tilde{U}(\theta)$, namely $A = 0$. In Figure 1 we plot the estimated density based on $10^6$ samples and the mean kinetic energy over iterations, when $\xi$ is fixed at $0.1, 1, 10$ successively, as well as when $\xi$ follows our thermostat update in (6).

From Figure 1, when $\xi = B = 1$, the SDE is the ordinary Langevin dynamics. In this case, the sampling is accurate and the kinetic energy is controlled around $0.5$. When $\xi > B$, the kinetic energy drops to a low value, and the sampling gets stuck in one local minimum; this is what happens in the SGD optimization with momentum. When $\xi < B$, the kinetic energy gets too high, and the sampling looks like a random walk. For SGNHT, the sampling looks as accurate as the one with $\xi = B$ and the kinetic energy is also controlled around $0.5$. Actually in Appendix B, we see that the value of $\xi$ of SGNHT quickly converges to $B = 1$.

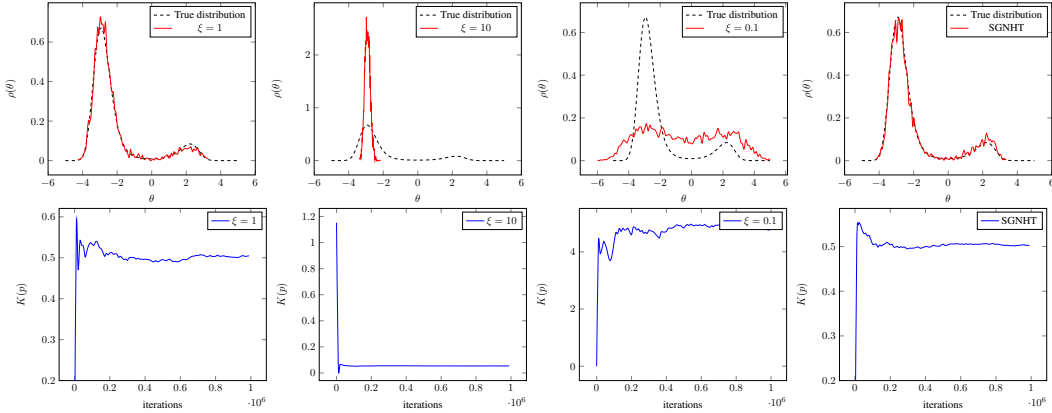

Figure 1: The samples on $\rho(\theta)$ and the mean kinetic energy over iterations $K(p)$ with $\xi = 1$ (1st), $\xi = 10$ (2nd), $\xi = 0.1$ (3rd), and the SGNHT (4th). The first three do not use a thermostat. The fourth column shows that the SGNHT method is able to sample accurately and maintains the mean kinetic energy with unknown noise.

## 4   The General Recipe

In this section, we mathematically justify the proposed SGNHT method. We begin with a theorem showing why and how a sampler based on SDEs using stochastic gradients can produce the correct target distribution. The theorem serves two purposes. First, one can examine whether a given SDE sampler is correct or not. The theorem is more general than previous ones in [5][24] which focus on justifying individual methods. Second, the theorem can be a general recipe for proposing new methods. As a concrete example of using this approach, we show how to obtain SGNHT from the main theorem.

### 4.1 The Main Theorem

Consider the following general stochastic differential equations that use the stochastic force:

$$d\mathbf{\Gamma} = \mathbf{v}(\mathbf{\Gamma})dt + \mathcal{N}(0, 2\,\mathbf{D}(\boldsymbol{\theta})dt) \tag{7}$$

where $\mathbf{\Gamma} = (\boldsymbol{\theta}, \mathbf{p}, \boldsymbol{\xi})$, and both $\mathbf{p}$ and $\boldsymbol{\xi}$ are optional. $\mathbf{v}$ is a vector field that characterizes the deterministic part of the dynamics. $\mathbf{D}(\boldsymbol{\theta}) = \mathbf{A} + diag(\mathbf{0}, \mathbf{B}(\boldsymbol{\theta}), \mathbf{0})$, where the injected noise $\mathbf{A}$ is known and constant, whereas the noise of the stochastic gradient $\mathbf{B}(\boldsymbol{\theta})$ is unknown, may vary, and only appears in blocks corresponding to rows of the momentum. Both $\mathbf{A}$ and $\mathbf{B}$ are symmetric positive semidefinite. Taking the dynamics of SGHMC as an example, it has $\mathbf{\Gamma} = (\boldsymbol{\theta}, \mathbf{p})$, $\mathbf{v} = (\mathbf{p}, \mathbf{f} - A\mathbf{p})$ and $\mathbf{D}(\boldsymbol{\theta}) = diag(\mathbf{0}, A\,\mathbf{I} - \hat{\mathbf{B}}(\boldsymbol{\theta}) + \mathbf{B}(\boldsymbol{\theta}))$.

Let $\rho(\mathbf{\Gamma}) = (1/Z)\exp(-H(\mathbf{\Gamma}))$ be the joint probability density of all variables, and write $H$ as $H(\mathbf{\Gamma}) = U(\boldsymbol{\theta}) + Q(\boldsymbol{\theta}, \mathbf{p}, \boldsymbol{\xi})$. The marginal density for $\boldsymbol{\theta}$ must equal the target density,

$$\exp\left(-U(\boldsymbol{\theta})\right) \propto \iint \exp\left(-U(\boldsymbol{\theta}) - Q(\boldsymbol{\theta}, \mathbf{p}, \boldsymbol{\xi})\right) d\mathbf{p}d\boldsymbol{\xi} \tag{8}$$

which will be referred as the marginalization condition.

**Main Theorem.** *The stochastic process of $\boldsymbol{\theta}$ generated by the stochastic differential equation* (7) *has the target distribution $\rho_\theta(\boldsymbol{\theta}) = (1/Z)\exp(-U(\boldsymbol{\theta}))$ as its stationary distribution, if $\rho \propto \exp(-H)$ satisfies the marginalization condition* (8)*, and*

$$\nabla \cdot (\rho\mathbf{v}) = \nabla\nabla^\top : (\rho\,\mathbf{D}), \tag{9}$$

*where we use concise notation, $\nabla = (\partial/\partial\boldsymbol{\theta}, \partial/\partial\mathbf{p}, \partial/\partial\boldsymbol{\xi})$ being a column vector,*

*$\cdot$ representing a vector inner product $\mathbf{x} \cdot \mathbf{y} = \mathbf{x}^\top\mathbf{y}$, and : representing a matrix double dot product $\mathbf{X} : \mathbf{Y} = \mathrm{trace}(\mathbf{X}^\top\mathbf{Y})$.*

*Proof.* See Appendix C. $\qquad\square$

**Remark.** *The theorem implies that when the SDE is solved exactly (namely $h \to 0$), then the noise of the stochastic force has no effect, because $\lim_{h\to0}\mathbf{D} = \mathbf{A}$ [5]. In this case, any dynamics that produce the correct distribution with the true gradient, such as the original Langevin dynamics, can also produce the correct distribution with the stochastic gradient.*

However, when there is discretization error one must find the proper $H$, $\mathbf{v}$ and $\mathbf{A}$ to ensure production of the correct distribution of $\boldsymbol{\theta}$. Towards this end, the theorem provides a general recipe for finding proper dynamics that can sample correctly in the presence of stochastic forces. To use this prescription, one may freely select the dynamics characterized by $\mathbf{v}$ and $\mathbf{A}$ as well as the joint stationary distribution for which the marginalization condition holds. Together, the selected $\mathbf{v}$, $\mathbf{A}$ and $\rho$ must satisfy this main theorem.

The marginalization condition is important because for some stochastic differential equations there exists a $\rho$ that makes (9) hold even though the marginalized distribution is not the target distribution. Therefore, care must be taken when designing the dynamics. In the following subsection, we will use the proposed stochastic gradient Nosé-Hoover thermostats as an illustrative example of how our recipe may be used to discover new methods. We will show more examples in Appendix D.

### 4.2 Revisiting the Stochastic Gradient Nosé-Hoover Thermostat

Let us start from the following dynamics:

$$d\boldsymbol{\theta} = \mathbf{p}\,dt, \quad d\mathbf{p} = \mathbf{f}dt - \mathbf{\Xi}\,\mathbf{p}\,dt + \mathcal{N}(0, 2\mathbf{D}\,dt),$$

where both $\mathbf{\Xi}$ and $\mathbf{D}$ are $n \times n$ matrices. Apparently, when $\mathbf{\Xi} \neq \mathbf{D}$, the dynamics will not generate the correct target distribution (see Appendix D). Now let us add dynamics for $\mathbf{\Xi}$, denoted by $d\mathbf{\Xi} = \mathbf{v}^{(\mathbf{\Xi})}\,dt$, and demonstrate application of the main theorem.

Let $\rho(\boldsymbol{\theta}, \mathbf{p}, \mathbf{\Xi}) = (1/Z)\exp(-H(\boldsymbol{\theta}, \mathbf{p}, \mathbf{\Xi}))$ be our target distribution, where $H(\boldsymbol{\theta}, \mathbf{p}, \mathbf{\Xi}) = U(\boldsymbol{\theta}) + Q(\mathbf{p}, \mathbf{\Xi})$ and $Q(\mathbf{p}, \mathbf{\Xi})$ is also to be determined. Clearly, the marginalization condition is satisfied for such $H(\boldsymbol{\theta}, \mathbf{p}, \mathbf{\Xi})$.

Let $R_{\mathbf{z}}$ denote the gradient of a function $R$, and $R_{\mathbf{z}\,\mathbf{z}}$ denote the Hessian. For simplicity, we constrain $\nabla_{\boldsymbol{\Xi}} \cdot \mathbf{v}^{(\boldsymbol{\Xi})} = 0$, and assume that $\mathbf{D}$ is a constant matrix. Then the LHS and RHS of (9) become

$$LHS = (\nabla \cdot \mathbf{v} - \nabla H \cdot \mathbf{v})\rho = (-\mathrm{trace}(\boldsymbol{\Xi}) + \mathbf{f}^{\mathsf{T}}\mathbf{p} - Q_{\mathbf{p}}^{\mathsf{T}}\mathbf{f} + Q_{\mathbf{p}}^{\mathsf{T}}\boldsymbol{\Xi}\mathbf{p} - Q_{\boldsymbol{\Xi}} : \mathbf{v}^{(\boldsymbol{\Xi})})\rho,$$

$$RHS = \mathbf{D} : \rho_{\mathbf{p}\mathbf{p}} = \mathbf{D} : (Q_{\mathbf{p}}Q_{\mathbf{p}}^{\mathsf{T}} - Q_{\mathbf{p}\mathbf{p}})\rho.$$

Equating both sides, one gets

$$-\mathrm{trace}(\boldsymbol{\Xi}) + \mathbf{f}^{\mathsf{T}}\mathbf{p} - Q_{\mathbf{p}}^{\mathsf{T}}\mathbf{f} + Q_{\mathbf{p}}^{\mathsf{T}}\boldsymbol{\Xi}\mathbf{p} - Q_{\boldsymbol{\Xi}} : \mathbf{v}^{(\boldsymbol{\Xi})} = \mathbf{D} : (Q_{\mathbf{p}}Q_{\mathbf{p}}^{\mathsf{T}}) - \mathbf{D} : Q_{\mathbf{p}\mathbf{p}}.$$

To cancel the $\mathbf{f}$ terms, set $Q_{\mathbf{p}} = \mathbf{p}$, then $Q(\mathbf{p}, \boldsymbol{\Xi}) = \frac{1}{2}\mathbf{p}^{T}\mathbf{p} + S(\boldsymbol{\Xi})$, which leaves $S(\boldsymbol{\Xi})$ to be determined. The equation becomes

$$-\boldsymbol{\Xi} : \mathbf{I} + \boldsymbol{\Xi} : (\mathbf{p}\mathbf{p}^{\mathsf{T}}) - S_{\boldsymbol{\Xi}} : \mathbf{v}^{(\boldsymbol{\Xi})} = \mathbf{D} : (\mathbf{p}\mathbf{p}^{\mathsf{T}}) - \mathbf{D} : \mathbf{I}. \tag{10}$$

Obviously, $\mathbf{v}^{(\boldsymbol{\Xi})}$ must be a function of $\mathbf{p}\mathbf{p}^{\mathsf{T}}$ since $S_{\boldsymbol{\Xi}}$ is independent of $\mathbf{p}$. Also, $\mathbf{D}$ must only appear in $S_{\boldsymbol{\Xi}}$, since we want $\mathbf{v}^{(\boldsymbol{\Xi})}$ to be independent of the unknown $\mathbf{D}$. Finally, $\mathbf{v}^{(\boldsymbol{\Xi})}$ should be independent of $\boldsymbol{\Xi}$, since we let $\nabla_{\boldsymbol{\Xi}} \cdot \mathbf{v}^{(\boldsymbol{\Xi})} = 0$. Combining all three observations, we let $\mathbf{v}^{(\boldsymbol{\Xi})}$ be a linear function of $\mathbf{p}\mathbf{p}^{\mathsf{T}}$, and $S_{\boldsymbol{\Xi}}$ a linear function of $\boldsymbol{\Xi}$. With some algebra, one finds that

$$\mathbf{v}^{(\boldsymbol{\Xi})} = (\mathbf{p}\mathbf{p}^{\mathsf{T}} - \mathbf{I})/\mu, \tag{11}$$

and $S_{\boldsymbol{\Xi}} = (\boldsymbol{\Xi} - \mathbf{D})\mu$ which means $Q(\mathbf{p}, \boldsymbol{\Xi}) = \frac{1}{2}\mathbf{p}^{\mathsf{T}}\mathbf{p} + \frac{1}{2}\mu(\boldsymbol{\Xi} - \mathbf{D}) : (\boldsymbol{\Xi} - \mathbf{D})$. (11) defines a general stochastic gradient Nosé-Hoover thermostats. When $\mathbf{D} = D\,\mathbf{I}$ and $\boldsymbol{\Xi} = \xi\,\mathbf{I}$ (here $D$ and $\xi$ are both scalars and $\mathbf{I}$ is the identity matrix), one can simplify (10) and obtain $v^{(\xi)} = (\mathbf{p}^{\mathsf{T}}\mathbf{p} - n)/\mu$. It reduces to (6) of the SGNHT in section 3 when $\mu = n$.

The Nosé-Hoover thermostat without stochastic terms has $\xi \sim \mathcal{N}(0, \mu^{-1})$. When there is a stochastic term $\mathcal{N}(0, 2\,\mathbf{D}\,dt)$, the distribution of $\boldsymbol{\Xi}$ changes to a matrix normal distribution $\mathcal{MN}(\mathbf{D}, \mu^{-1}\,\mathbf{I}, \mathbf{I})$ (in the scalar case, $\mathcal{N}(D, \mu^{-1})$). This indicates that the thermostat *absorbs* the stochastic term $\mathbf{D}$, since the expected value of $\boldsymbol{\Xi}$ is equal to $\mathbf{D}$, and leaves the marginal distribution of $\boldsymbol{\theta}$ invariant.

In the derivation above, we assumed that $\mathbf{D}$ is constant (by assuming $\mathbf{B}$ constant). This assumption is reasonable when the data size is large so that the posterior of $\boldsymbol{\theta}$ has small variance. In addition, the full dynamics of $\boldsymbol{\Xi}$ requires additional $n \times n$ equations of motion, which is generally too costly. In practice, we found that Algorithm 1 with a single scalar $\xi$ works well.

## 5 Experiments

### 5.1 Gaussian Distribution Estimation Using Stochastic Gradient

We first demonstrate our method on a simple example: Bayesian inference on 1D normal distributions. The first part of the experiment tries to estimate the mean of the normal distribution with known variance and $N = 100$ random examples from $\mathcal{N}(0, 1)$. The likelihood is $\mathcal{N}(x_i|\mu, 1)$, and an improper prior of $\mu$ being uniform is assigned. Each iteration we randomly select $\tilde{N} = 10$ examples. The noise of the stochastic gradient is a constant given $\tilde{N}$ (Appendix E).

Figure 2 shows the density of $10^6$ samples obtained by SGNHT (1st plot) and SGHMC (2nd plot). As we can see, SGNHT samples accurately without knowing the variance of the noise of the stochastic force under all parameter settings, whereas SGHMC samples accurately only when $h$ is small and $A$ is large. The 3rd plot shows the mean of $\xi$ values in SGNHT. When $h = 0.001$, $\xi$ and $A$ are close. However, when $h = 0.01$, $\xi$ becomes much larger than $A$. This indicates that the discretization introduces a large noise from the stochastic gradient, and the $\xi$ variable effectively absorbs the noise.

The second part of the experiment is to estimate both mean and variance of the normal distribution. We use the likelihood function $\mathcal{N}(x_i|\mu, \gamma^{-1})$ and the Normal-Gamma distribution $\mu, \gamma \sim \mathcal{N}(\mu|0, \gamma)\mathrm{Gam}(\gamma|1, 1)$ as prior. The variance of the stochastic gradient noise is no longer a constant and depends on the values of $\mu$ and $\gamma$ (see Appendix E).

Similar density plots are available in Appendix E. Here we plot the Root Mean Square Error (RMSE) of the density estimation vs. the autocorrelation time of the observable $\mu + \gamma$ under various $h$ and $A$ in the 4th plot in Figure 2. We can see that SGNHT has significantly lower autocorrelation time than SGHMC at similar sampling accuracy. More details about the $h$, $A$ values which produces the plot are also available in Appendix E.

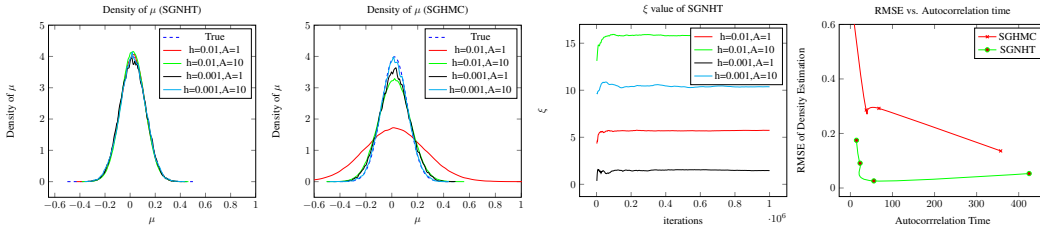

Figure 2: Density of $\mu$ obtained by SGNHT with known variance (1st), density of $\mu$ obtained by SGHMC with known variance (2nd), mean of $\xi$ over iterations with known variance in SGNHT (3rd), RMSE vs. Autocorrelation time for both methods with unknown variance (4th).

## 5.2 Machine Learning Applications

In the following machine learning experiments, we used a reformulation of (5) and (6) similar to [5], by letting $\mathbf{u} = \mathbf{p}\,h$, $\eta = h^2$, $\alpha = \xi h$ and $a = Ah$. The resulting Algorithm 2 is provided in Appendix F. In [5], SGHMC has been extensively compared with SGLD, SGD and SGD-momentum. Our experiments will focus on comparing SGHMC and SGNHT. Details of the experiment settings are described below. The test results over various parameters are reported in Figure 3.

**Bayesian Neural Network** We first evaluate the benchmark MNIST dataset, using the Bayesian Neural Network (BNN) as in [5]. The MNIST dataset contains 50,000 training examples, 10,000 validation examples, and 10,000 test examples. To show our algorithm being able to handle large stochastic gradient noise due to small minibatch, we chose the minibatch of size 20. Each algorithm is run for a total number of 50k iterations with burn-in of the first 10k iterations. The hidden layer size is 100, parameter $a$ is from $\{0.001, 0.01\}$ and $\eta$ from $\{2, 4, 6, 8\} \times 10^{-7}$.

**Bayesian Matrix Factorization** Next, we evaluate our methods on two collaborative filtering tasks: the Movielens ml-1m dataset and the Netflix dataset, using the Bayesian probabilistic matrix factorization (BPMF) model [21]. The Movielens dataset contains 6,050 users and 3,883 movies with about 1M ratings, and the Netflix dataset contains 480,046 users and 17,000 movies with about 100M ratings. To conduct the experiments, Each dataset is partitioned into training (80%) and testing (20%), and the training set is further partitioned for 5-fold cross validation. Each minibatch contains 400 ratings for Movielens1M and 40k ratings for Netflix. Each algorithm is run for 100k iterations with burn-in of the first 20k iterations. The base number is chosen as 10, parameter $a$ is from $\{0.01, 0.1\}$ and $\eta$ from $\{2, 4, 6, 8\} \times 10^{-7}$.

**Latent Dirichlet Allocation** Finally, we evaluate our method on the ICML dataset using Latent Dirichlet Allocation [4]. The ICML dataset contains 765 documents from the abstracts or ICML proceedings from 2007 to 2011. After simple stopword removal, we obtained a vocabulary size of about 2K and total words of about 44K. We used 80% documents for 5-fold cross validation and the remaining 20% for testing. Similar to [18], we used the semi-collapsed LDA whose posterior of $\theta_{kw}$ is provided in Appendix H. The Dirichlet prior parameter for the topic distribution for each document is set to 0.1 and the Gaussian prior for $\theta_{kw}$ is set as $\mathcal{N}(0.1, 1)$. Each minibatch contains 100 documents. Each algorithm is run for 50k iterations with the first 10k iterations as burn-in. Topic number is 30, parameter $a$ is from $\{0.01, 0.1\}$ and $\eta$ from $\{2, 4, 6, 8\} \times 10^{-5}$.

### 5.2.1 Result Analysis

From Figure 3, SGNHT is apparently more stable than SGHMC when the discretization step $\eta$ is larger. In all four datasets, especially with the smaller $a$, SGHMC gets worse and worse results as $\eta$ increases. With the largest $\eta$, SGHMC diverges (as the green curve is way beyond the range) due to its failure to handle the large unknown noise with small $a$.

Figure 3 also gives a comprehensive view of the critical role that $a$ plays on. On one hand, larger $a$ may cause more random walk effect which slows down the convergence (as in Movielens1M and Netflix). On the other hand, it is helpful to increase the ergodicity and compensate the unknown noise from the stochastic gradient (as in MNIST and ICML).

Throughout the experiment, we find that the kinetic energy of SGNHT is always maintained around 0.5 while that of SGHMC is usually higher. And overall SGNHT has better test performance with the choice of the parameters selected by cross validation (see Table 2 of Appendix G).

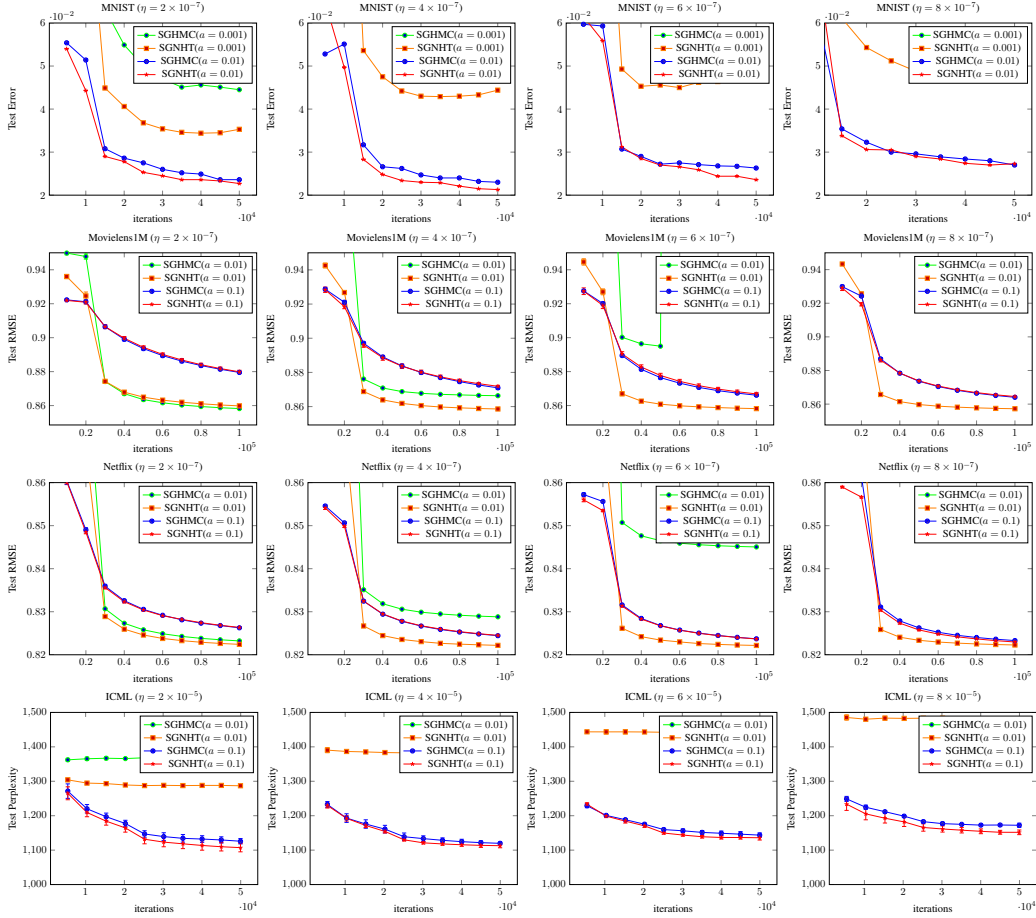

Figure 3: The test error of MNIST (1st row), test RMSE of Movielens1M (2nd row), test RMSE of Netflix (3rd row) and test perplexity of ICML (4th row) datasets with their standard deviations (close to 0 in row 2 and 3) under various $\eta$ and $a$.

## 6 Conclusion and Discussion

In this paper, we find proper dynamics that adpatively fit to the noise introduced by stochastic gradients. Experiments show that our method is able to control the temperature, estimate the unknown noise, and perform competitively in practice. Our method can be justified in continuous time by a general theorem. The discretization of continuous SDEs, however, introduces bias. This issue has been extensively studied by previous work such as [20, 22, 15, 12]. The existency of an invariant measure has been proved (e.g., Theorem 3.2 [22] and Proposition 2.5 [12]) and the bound of the error has been obtained (e.g, $\mathcal{O}(h^2)$ for a symmetric splitting scheme [12]). Due to space limitation, we leave a deeper discussion on this topic and a more rigorous justification to future work.

### Acknowledgments

We acknowledge Kevin P. Murphy and Julien Cornebise for helpful discussions and comments.

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
