[Supplementary Material]

# Supplementary Material for: Bayesian Sampling Using Stochastic Gradient Thermostats

## A  Additional Notes on Dynamics-Based Methods

### A.1  Hybrid Monte Carlo

The procedure of generating a sample in HMC can be described as follows: (Assume the last iteration sample is $\boldsymbol{\theta}$)

1. Generate $\mathbf{p}$ from a standard normal distribution;
2. Compute $(\boldsymbol{\theta}', \mathbf{p}') = \Psi_\tau(\boldsymbol{\theta}, \mathbf{p})$, where $\Psi_\tau$ is the composition of $\nu$ numerical integrators $\Psi_{\Delta t}$ (e.g., the "leapfrog" method) of the Hamiltonian system, namely $\Psi_\tau = \Psi_{\Delta t} \circ \Psi_{\Delta t} \circ ... \circ \Psi_{\Delta t}$ and $\Delta t = \tau/\nu$;
3. Accept $\boldsymbol{\theta}'$ with probability $\min\{1, \rho(\boldsymbol{\theta}', \mathbf{p}')/\rho(\boldsymbol{\theta}, \mathbf{p})\}$ and stay with $\boldsymbol{\theta}$ otherwise.

### A.2  Wiener Process

The Langevin dynamics is described by the following SDE:
$$\mathrm{d}\boldsymbol{\theta} = \mathbf{p}\,\mathrm{d}t,$$
$$\mathrm{d}\,\mathbf{p} = \mathbf{f}(\boldsymbol{\theta})\mathrm{d}t - A\,\mathbf{p}\,\mathrm{d}t + \sqrt{2A}\mathrm{d}\,\mathbf{W},$$
where $A$ is the damping constant and $\mathbf{W}$ is $n$ independent Wiener processes. A Wiener process $W$ satisfies:

1. $W(0) = 0$ with probability 1;
2. $W(t+h) - W(t) \sim \mathcal{N}(0, h)$ and is independent of $W(\tau)$ for $\tau \leq t$.

Since the increment of $W$ is a normal distribution, $dW$ is informally written as $\mathcal{N}(0, dt)$ in [5]. Strictly speaking, $W$ is a continuous function of $t$, whereas $\mathcal{N}(0, dt)$ is a random variable independent of $t$ such that it is reluctant to be put in a continuous SDE. But in this paper, we still follow [5] for readers who are not familiar with SDE.

## B  Additional Plots of the Double-Well Potential Illustration

See the plot of $\xi$ value of SGNHT in Figure 4.

Figure 4: SGNHT is able to automatically adapt $\xi$ to $B = 1$.

## C  The Proof of the Main Theorem

*Proof.* We first show that the stationary probability density of the stochastic differential equation is $\rho$. The probability density of $\boldsymbol{\Gamma}$, $\rho'(\boldsymbol{\Gamma}, t)$ satisfies the Fokker-Planck equation:

$$\frac{\partial \rho'(\boldsymbol{\Gamma}, \mathbf{t})}{\partial t} + \nabla_{\boldsymbol{\Gamma}} \cdot (\rho'(\boldsymbol{\Gamma}, t)f(\boldsymbol{\Gamma})) - \nabla_{\Gamma}\nabla_{\Gamma}^T : (\rho'(\boldsymbol{\Gamma}, t)D(\boldsymbol{\Gamma})) = 0. \tag{12}$$

For the stationary probability density,

$$\frac{\partial \rho'(\mathbf{\Gamma}, t)}{\partial t} = 0.$$

So we have

$$\nabla_\Gamma \cdot (\rho'(\mathbf{\Gamma})f(\mathbf{\Gamma})) - \nabla_\Gamma \nabla_\Gamma^T : (\rho'(\mathbf{\Gamma})D(\mathbf{\Gamma})) = 0.$$

This means that as long as $\rho$ satisfies (9), it can be preserved by the dynamics. And because $\rho$ satisfies the marginalization condition, the marginal density of $\boldsymbol{\theta}$, which equals the target density, is also preserved by the dynamics. This completes the proof.

$\square$

## D  Additional Examples of Applying the Main Theorem

This section presents some additional examples. For simplicity, we only show the 1-d case.

### D.1  Brownian Dynamics

In Welling's [24], the sampler is Brownian dynamics with stochastic force:

$$\theta dt = f dt + \sqrt{2(1 + B)} dW. \tag{13}$$

It is easy to obtain from the Fokker-Planck equation that the stationary distribution is

$$\rho(\theta) = \frac{1}{Z} \exp(-(1 + B)^{-1} U(\theta)). \tag{14}$$

Namely, the sampler samples from the target distribution with an increased temperature. There are two easy ways to eliminate the effect of $B$. One is to modify the random term to be $\sqrt{2(1 + B - \hat{B})} dW$, and the other is to modify the deterministic part to be $(1 + \hat{B})f dt$. When $\hat{B} = B$, the sampler produces the correct distribution.

### D.2  Langevin Dynamics

The Langevin dynamics with stochastic force

$$d\theta = p dt, \tag{15}$$
$$dp = f dt - Ap dt + \sqrt{2(A + B)} dW, \tag{16}$$

also has an additional noise term $B$. It is easy to see from the theorem that this dynamics does not produce the correct distribution when $B \neq 0$. In [5], this is corrected by modifying the random term to be $\sqrt{2(A + B - \hat{B})} dW$. Another way to correct it is to change $Ap dt$ to $(A + \hat{B})p dt$. When $\hat{B} = B$, all modification will make the sampling correct.

In practice, however, it is usually hard to have a good estimation of $B$. By our observation, the value of $B$ not only depends on the variance of the stochastic force noise, but also depends on the discretization method. So it can be written as a function of $V$ and all parameters in the dynamics, i.e., $B(V, h, A)$. It satisfies that $B \to 0$ as $h \to 0$, and empirically we found that $B \propto \mathcal{O}(h^\gamma)$, where $\gamma$ is between 1.5 and 2. Although the theorem does not change with this $B$, it does create some practical difficulties. Therefore, a method that does not need to estimate $B$ becomes more interesting.

## E  Additional Details about the Gaussian Distribution Estimation Experiment

Our first experiment is to estimate the mean of a normal distribution with known variance: given $N$ i.i.d. examples $x_i \sim \mathcal{N}(\mu, 1)$, and an improper prior of $\mu$ being uniform, we have the posterior of

$\mu \sim \mathcal{N}(\bar{x}, 1/N)$, where $\bar{x} = \sum_{i=1}^{N} x_i/N$. The stochastic gradient of the log density is

$$\nabla \tilde{U} = N\mu - \frac{N}{\tilde{N}} \sum_{i=1}^{\tilde{N}} x_i.$$

The noise of the stochastic gradient is independent of $\mu$, so it is constant given $\tilde{N}$. Figure 5 shows that SGHMC (solid lines) does not have correct kinetic energy unless $h$ small and $A$ large.

The second experiment is to estimate both the mean and the variance of a normal distribution: given $N$ i.i.d. examples $x_i \sim \mathcal{N}(\mu, \gamma^{-1})$, and a Normal-Gamma distribution $\mu, \gamma \sim \mathcal{N}(\mu|0, \gamma^{-1})\mathrm{Gam}(\gamma|1, 1)$ as the prior, we obtain a posterior which is another Normal-Gamma distribution $p(\mu, \gamma|D) \propto \sim \mathcal{N}(\mu|\mu_N, (\kappa_N\gamma)^{-1})\mathrm{Gam}(\gamma|\alpha_N, \beta_N)$, where

$$\mu_N = \frac{N}{N+1}\bar{x}, \quad \kappa_N = 1 + N, \quad \alpha_N = 1 + \frac{N}{2},$$

$$\beta_N = 1 + \frac{1}{2}\sum_{i=1}^{N}(x_i - \bar{x})^2 + \frac{N}{2(1+N)}(\bar{x} - \mu_0)^2.$$

The stochastic gradient is

$$\nabla_\mu \tilde{U} = (N+1)\mu\gamma - \frac{N}{\tilde{N}}\gamma \sum_{i=1}^{\tilde{N}} x_i,$$

$$\nabla_\gamma \tilde{U} = -(N+1)/(2\gamma) + 1 + 1/2\mu^2 + \frac{N}{2\tilde{N}}\sum_{i=1}^{\tilde{N}}(x_i - \mu)^2.$$

It can be seen that the variance of the noise depends on the values of $\mu$ and $\gamma$.

Figure 6 shows the marginal posterior density of $\mu$ and $\gamma$ of $10^6$ samples. It can be seen that SGNHT generates reasonably good samples with various $h$ and $A$. SGHMC, on the other hand, only works when $h$ is small and $A$ is large.

Table 1 shows the Root Mean Square Error (RMSE) of the density estimation and the autocorrelation time ($\tau$) of the observable $\mu + \gamma$ under various $h$ and $A$. We see that one should prefer large $h$ and small $A$ in order to sample efficiently, but small $h$ and large $A$ to be more accurate. SGNHT is overall more accurate and more efficient than SGHMC, and also more robust to the parameters choices for getting both accurate sampling and low autocorrelation time.

Figure 5: 1D normal distribution with unknown mean $\mu$ and known variance (kinetic energy).

Table 1: (RMSE, Autocorrelation time) of SGHMC and SGNHT on the synthetic dataset.

| | $h = 0.01, A = 1$ | $h = 0.01, A = 10$ | $h = 0.001, A = 1$ | $h = 0.001, A = 10$ |
|---|---|---|---|---|
| SGHMC | (0.629,7.19) | (0.286,37.68) | (0.292,67.83) | (0.136,357.14) |
| SGNHT | (0.175,14.35) | (0.026,55.65) | (0.091,22.97) | (0.053,424.81) |

## F   The Reformulated SGNHT Algorithm

The reformulated SGNHT algorithm with $\mathbf{u} = \mathbf{p}h$, $\eta = h^2$, $\alpha = \xi h$ and $a = Ah$ is in Algorithm 2.

Figure 6: 1D Normal with unknown $\mu$ and $\gamma$

---

**Algorithm 2:** Reformulated Stochastic Gradient Nosé-Hoover Thermostat

---

**Input**: Parameters $\eta$, $a$.

Initialize $\boldsymbol{\theta}_{(0)} \in \mathbf{R}^n$, $\mathbf{u}_{(0)} \sim \mathcal{N}(0, \eta\,\mathbf{I})$, and $\alpha_{(0)} = a$ ;

**for** $t = 1, 2, \ldots$ **do**

    Evaluate $\nabla \tilde{U}(\boldsymbol{\theta}_{(t-1)})$ from (2) ;

    $\mathbf{u}_{(t)} = \mathbf{u}_{(t-1)} - \alpha_{(t-1)}\,\mathbf{u}_{(t-1)} - \nabla \tilde{U}(\boldsymbol{\theta}_{(t-1)})\eta + \mathcal{N}(0, 2a\eta)$;

    $\boldsymbol{\theta}_{(t)} = \boldsymbol{\theta}_{(t-1)} + \mathbf{u}_{(t)}$;

    $\alpha_{(t)} = \alpha_{(t-1)} + (\frac{1}{n}\,\mathbf{u}_{(t)}^{\top}\,\mathbf{u}_{(t)} - \eta)$;

**end**

---

## G    Additional Results on the Machine Learning Experiments

Supplementary experimental results on machine learning tasks including the chosen parameters based on validation set as well as their test error are available in Table 2 and Table 3.

Table 2: Test performances of SGHMC and SGNHT on the machine learning tasks (MNIST: test error; Movielens1M: tRMSE; Netflix: tRMSE; ICML-Abstract: test perplexity). The parameters for each algorithm were chosen based on the best performances on the validation sets.

|       | MNIST | Movielens1M | Netflix | ICML |
|-------|-------|-------------|---------|------|
| SGHMC | 2.25  | $0.8582 \pm 0.0004$ | $0.8232 \pm 0.0000$ | $1125.2 \pm 8.9$ |
| SGNHT | **2.15** | $\mathbf{0.8572 \pm 0.0003}$ | $\mathbf{0.8224 \pm 0.0002}$ | $\mathbf{1102.7 \pm 6.7}$ |

## H    The Posterior of the Latent Dirichlet Allocation

Let $\mathbf{W} = \{w_{d\ell}\}$ be the observed words, $\mathbf{Z} = \{z_{d\ell}\}$ be the topic indicator variables, where $d$ indexes the documents and $\ell$ indexes the words. Let $(\pi)_{kw}$ be the topic-word distribution, $n_{dkw}$ be the number of word $w$ in document $d$ allocated to topic $k$, $\cdot$ means marginal sum, i.e. $n_{dk\cdot} = \sum_w n_{dkw}$. The semi-collapsed posterior of the LDA model is:

$$p(\mathbf{W}, \mathbf{Z}, \pi | \alpha, \tau) = p(\pi | \tau) \prod_{d=1}^{D} p(\vec{w}_d, \vec{z}_d | \alpha, \pi),$$

where $D$ is the number of documents, $\alpha$ is the parameter in the Dirichlet prior of the topic distribution for each document, $\tau$ is the parameter in the prior of $\pi$, and

$$p(\vec{w}_d, \vec{z}_d | \alpha, \pi) = \prod_{k=1}^{K} \frac{\Gamma(\alpha + n_{dk\cdot})}{\Gamma(\alpha)} \prod_{w=1}^{W} \pi_{kw}^{n_{dkw}}.$$

With the Expanded-Natural representation of the simplexes $\pi_k$'s in [18], where

$$\pi_{kw} = \frac{e^{\theta_{kw}}}{\sum_{w'} e^{\theta_{kw'}}},$$

and a Gaussian prior on $\theta_{kw}$,

$$p(\theta_{kw} | \tau = \{\beta, \sigma\}) = \frac{1}{2\pi\sigma} e^{-\frac{(\theta_{kw} - \beta)^2}{2\sigma^2}},$$

Table 3: The chosen parameters $(a, \eta)$ of SGHMC and SGNHT based on the validation performance on the machine learning tasks.

| | MNIST | Movielens1M | Netflix | ICML |
|---|---|---|---|---|
| SGHMC | $(0.01, 2 \times 10^{-7})$ | $(0.01, 2 \times 10^{-7})$ | $(0.01, 2 \times 10^{-7})$ | $(0.1, 2 \times 10^{-5})$ |
| SGNHT | $(0.01, 2 \times 10^{-7})$ | $(0.01, 8 \times 10^{-7})$ | $(0.01, 8 \times 10^{-7})$ | $(0.1, 2 \times 10^{-5})$ |

the stochastic gradient of the log-posterior of parameter $\theta_{kw}$ with a mini-batch of size $D_t$ becomes,

$$\frac{\partial \tilde{U}(\boldsymbol{\theta})}{\partial \theta_{kw}} = \frac{\partial \log \tilde{p}(\boldsymbol{\theta} | \mathbf{W}, \tau, \alpha)}{\partial \theta_{kw}} = \frac{\beta - \theta_{kw}}{\sigma^2} + \frac{D}{D_t} \sum_{d \in \text{batch}} \mathbb{E}_{\vec{z}_d | \vec{w}_d, \theta, \alpha} \left( n_{dkw} - \pi_{kw} n_{dk\cdot} \right).$$

When $\sigma = 1$, we have the same stochastic gradient as in [18]. To calculate the expectation term, we use the same Gibbs sampling method as in [18],

$$p(z_{d\ell} = k | \vec{w}_d, \theta, \alpha) = \frac{\left( \alpha + n_{dk\cdot}^{\backslash \ell} \right) e^{\theta_{kw_{d\ell}}}}{\sum_{k'} \left( \alpha + n_{dk'\cdot}^{\backslash \ell} \right) e^{\theta_{k'w_{d\ell}}}},$$

where $\backslash \ell$ denotes the count excluding the $\ell$-th topic assignment variable. The expectation is estimated by the samples.