[Reviews · NeurIPS 2014]

Submitted by Assigned_Reviewer_13

The paper is concerned with Monte Carlo sampling based on the discretisation of SDEs. This is a particularly topical subject since there has been some interest lately in such techniques due to the fact that they allow for the use of "stochastic gradients" which are particularly appealing in some "big data" settings since they allow one to run algorithms with only partial evaluation of the likelihood/energy function.

The paper is particularly well written and pedagogical. In additional it clarifies earlier contributions and provides a rigorous overview of the main results useful in this emerging area. In particular the authors propose a Nose-Hoover based sampling algorithm, which is guaranteed to produce samples from the correct distribution (before discretisation naturally), which is in contrast with earlier contributions. This will be useful to people, like me, who are not very familiar with the molecular simulation literature.

This is an excellent paper, likely to have impact.

Summary: Explores ways of using discretised SDEs in order to produce samples guaranteed to be approximately sampled from a distribution of interest in the situation where one wants to avoid evaluating the posterior density / energy function at each iteration.

Submitted by Assigned_Reviewer_25

This is an interesting paper that exploits the NH thermostat to fix the problem of potentially severe bias and non-existence of an invariant measure in sampling schemes comprised of stochastic gradient estimates.

The paper is nicely written and the empirical evaluation is comprehensive.

A nonreversible hypo elliptic SDE is used as the basis of this work. As soon as the grad of the potential is replaced with a state dependent stochastic process (the SG in this case) - then there is no guarantee of the existence of an invariant measure or indeed that the SDE is non-explosive. The limiting behaviours are only valid in the limits of discrete form converging to the limiting process. This is a very shaky basis upon which to develop any type of convergent stable algorithm amenable to formal analysis and subsequent development and practical usage in serious applications.

Given the fundamental theoretical issues with such SG type schemes it is imo inadvisable to consider further developments in this vein. It should be noted that correct algorithms have appeared such as the Firefly methodology so one wonders why one should wish to persist in following a seam of research that at its foundation is fundamentally flawed. Of course a proof of correctness under necessary and sufficient conditions would change this sceptical perspective and would then be an important contribution.
Summary: nicely written paper.
good evaluation.
flawed methodology.
alternative **exact** schemes with proper working theoretical guarantees (Maclaurin and adams) are now appearing.

Submitted by Assigned_Reviewer_31

This paper introduces a family of stochastic processes for purposes of MCMC, in which an extra term is introduced to an SDE (eqs. 5-6), so that deviation from a target value of kinetic energy is penalized. The paper is of fair quality but I found it rather vague in places and I was not able to extract the precise essence of the method. In particular I was not able to understand the following:

1) line 55. This seems key, but I did not find a clear explanation of what this means: "To approximate a canonical ensemble, one critical condition that has been overlooked by existing methods is that, the system temperature (proportional to the mean kinetic
energy) must remain near a target temperature."

2) line 86 I don't know what re-randomise or correct mean here "One has to re-randomize p according to the standard normal distribution to get the correct canonical distribution."

3) line 96 I don't know what this means "Brownian dynamics.....is obtained from Langevin dynamics by rescaling time..."

4) line 133 equation (4). This seems to be at the heart of the paper, but was does the approximation sign here really mean? Moreover, on line 139 below it is stated: "The condition itself may look a bit counterintuitive,
but it is actually a necessary condition for p being distributed as its marginal canonical
distribution" What does this mean? What precisely is meant by equation (4)?

I found the paper too vague to be understandable. There may be some interesting content, but I am not sure what the motivation for the proposed method is. The authors mention line 154: "(6) appears to be the same as the Nos´e-Hoover thermostat". This is a tantalising connection, but without more clarity I can't be sure about the originality or the significance of this paper.
Summary: Some potentially interesting ideas from statistical physics are presented but the paper is not clear enough for me to understood the motivation for the proposed method.

Submitted by Meta_Reviewer_5

The Firefly paper is interesting, but to me it looked like a bunch of
work for a tiny speed-up: I wouldn't recommend practitioners bother. It
isn't clear that developments in that direction will be useful either,
so it's not time to shut down brainstorming of other ideas.

Welling and Teh's work has had impact. I don't think it's particularly
justified except in the limit of small step sizes either? The
justification of past work has largely been empirical. I think this
paper's three varied and moderate-sized applications is good enough.
The benefits aren't huge, but they are there.

Negatives:

While I can infer what is meant in some of the places that
Assigned_Reviewer_31 queries, I agree that the paper is hard to follow.
The core idea is barely explained at all. The background section and the
material around equation (4) would benefit from revision.

Given the exposition in the paper, it's perhaps surprising that the 1D
example (n=1) works out so well, given that above (4) it says that n is
large. However, in this example, things are favourable because having a
scalar \xi is ok. I'm more worried about the high-dimensional case, when
sampling noise can upset the momentum in different directions
differently. Then fixing up the scalar p'*p/n isn't enough to sample
from the correct distribution. (Acknowledged around the end of Section
4.2, but could be made clearer / more prominent in discussion.)

Final opinion:

It's an interesting approach, with some
weaknesses in the paper. The sampler won't be correct, but the
experimental evaluation shows incremental improvement over past work.

The dynamics: The presentation of SDE's is hard going, and could be
improved with a discussion of the stability of the discrete-time
algorithm. I don't think their potentially-unstable choice of discrete
dynamics is necessarily fatal if it works (although a discussion would
be nice). I'm personally more concerned theoretically with the posterior
being wrong due to non-isotropic noise.
Summary: As a reviewer, I'd annoyingly give this a 6. I don't think it would be
bad to include. Although revisions could make the paper better, and
there may be better more-polished papers that should beat it out.
Author Feedback
Author rebuttal: We thank the reviewers for their valuable comments.

Reviewer #25:

1. On the existence of the invariant density for SGNHT after discretization:

The underlying SDE is the same for the stochastic gradient as it is for the full gradient: the state-dependent "friction" tensor $B(theta)$ from SG is proportional to the step size $h$. So the issue is one of discretization error. Discretized SDEs are the method of choice for sampling in studies of soft matter. Only now are rigorous results emerging that theoretically justify discretized Langevin dynamics [1,2]. Although it is not the aim of the present paper to provide rigorous proofs, we provide some preliminary analysis as follows.

First of all, the extra diffusion term $B(\theta)$ from the stochastic gradient does not change the invariant density $\rho$ of the continuous dynamics. Only after discretization, the relation of diffusion and the variance of the SG is $B(\theta) = V(\theta) h/2$ (shown in Appendix B), where $V(\theta)$ is the variance of the SG and $h$ is the step size.

When the likelihood is from an exponential family whose log-likelihood is of the form $<\theta,\phi(x)> - G(\theta)$, $V(\theta)$ is independent of the state $\theta$. In this case, the SG based SDE is still state independent and SGNHT has an invariant density whose marginal is equal to the target density (shown in Section 4.2). After proper discretization such as splitting method, the existence of a modified invariant density can be proved (as in [2]).

When $V(\theta)$ is not a constant, the diffusion becomes state dependent after discretization. For the SGNHT, we have verified the existence of the leading term $g$ in an expansion of a modified invariant density $\tilde{\rho} = (1 + gh + O(h^2))\rho$ based on similar methodology as in [1]. Based on this result, it would be surprising if the SG version of the discretized SDEs cannot be justified for an interesting class of applications (and it has been justified empirically).

2. On the Firefly MC:
Although the Firefly MC provides an exact subset sampling scheme, practically SG based sampling methods are significantly more flexible and scalable than the Firefly MC. First of all, Firefly MC does not work well with high-dimensional multimodal distributions because a good lower bound function is hard to find. This makes it less efficient in large scale machine learning problems including the four applications in our paper. Secondly, SG based methods can explicitly tune the speedup by choosing different minibatch sizes, while Firefly MC cannot. In our experiment, the speedup of SGNHT is over 2000 compared to full-batch HMC, while the speedup of the Firefly MC is less than 30 in its experiments (and would be even less in our applications).

Reviewer #31:

We should offer revisions that clarify these points:

1) It means Eq(4). We will discuss the issue with Eq(4) in 4).
In statistical physics, $kT$ characterizes the target temperature, and $p^T p/n$ measures an instantaneous temperature (system temperature is defined as an average).

2) This is one of the two main steps of the HMC algorithm. Please refer to [3] (page 12) for more details. This step is not required in our method.

3) Please refer to [1] (page 3) for details. This part is also not directly related to our method.

4) Eq(4) is not used in our algorithms or theorems, but serves as an intuitive motivation of Eq(6). A better way to write would be,
$$ kT = \E [p^T p/n], $$
where $\E$ means the expectation.

Originality:
Our algorithm can be derived from Eq(4) (physically) or Eq(9) (mathematically, and the result is more general). The simple form of the general dynamics is described in Algorithm 1 and it is recommended to be used in practice for its simplicity. The simple algorithm rediscovered Eq(6) which appears to be the same as the equation of $\xi$ in NHT known in statistical mechanics. However, there are two main differences: 1. NHT is deterministic whereas we introduced the diffusion term in the $p$ equation to make it a SDE and also used the stochastic force; 2. NHT does not have the general full matrix form. We are the first to introduce the thermostat in stochastic gradient based sampling and apply it in large scale machine learning applications. We also built a framework to discover and justify new SDE based methods with SG.

References:

[1] “Rational construction of stochastic numerical methods for molecular sampling”,
Leimkuhler & Matthews (2012)

[2] “The computation of averages from equilibrium and nonequilibrium Langevin molecular dynamics”,
Leimkuhler, Matthews and Stoltz(2014)

[3] “Hamiltonian Monte Carlo”, Neal (2012)